# The Cytokinins BAP and 2-iP Modulate Different Molecular Mechanisms on Shoot Proliferation and Root Development in Lemongrass (*Cymbopogon citratus*)

**DOI:** 10.3390/plants12203637

**Published:** 2023-10-21

**Authors:** María del Rosario Cárdenas-Aquino, Alberto Camas-Reyes, Eliana Valencia-Lozano, Lorena López-Sánchez, Agustino Martínez-Antonio, José Luis Cabrera-Ponce

**Affiliations:** 1Departamento de Ingeniería Genética, Cinvestav Irapuato, Km. 9.6 Libramiento Norte Carr. Irapuato-León, Irapuato Gto 36824, Mexico; mrca85@aol.com (M.d.R.C.-A.); alberto.camas@cinvestav.mx (A.C.-R.); eliana.valencia@cinvestav.mx (E.V.-L.); 2Red de Estudios Moleculares Avanzados, Unidad de Microscopia Avanzada, Instituto de Ecología, A.C. INECOL 1975–2023, Carretera antigua a Coatepec 351, Col. El Haya, Xalapa 91073, Mexico; lorena.lopez@inecol.mx

**Keywords:** *Cymbopogon citratus*, lemongrass, shoot proliferation, root induction cytokinins (CKs), 2-iP, BAP, gene expression analysis

## Abstract

The known activities of cytokinins (CKs) are promoting shoot multiplication, root growth inhibition, and delaying senescence. 6-Benzylaminopurine (BAP) has been the most effective CK to induce shoot proliferation in cereal and grasses. Previously, we reported that in lemongrass (*Cymbopogon citratus*) micropropagation, BAP 10 µM induces high shoot proliferation, while the natural CK 6-(γ,γ-Dimethylallylamino)purine (2-iP) 10 µM shows less pronounced effects and developed rooting. To understand the molecular mechanisms involved, we perform a protein–protein interaction (PPI) network based on the genes of *Brachypodium distachyon* involved in shoot proliferation/repression, cell cycle, stem cell maintenance, auxin response factors, and CK signaling to analyze the molecular mechanisms in BAP versus 2-iP plants. A different pattern of gene expression was observed between BAP- versus 2-iP-treated plants. In shoots derived from BAP, we found upregulated genes that have already been demonstrated to be involved in de novo shoot proliferation development in several plant species; CK receptors *(AHK3, ARR1)*, stem cell maintenance *(STM*, *REV* and *CLV3*), cell cycle regulation (*CDKA-CYCD3* complex), as well as the auxin response factor (*ARF5*) and CK metabolism (*CKX1*). In contrast, in the 2-iP culture medium, there was an upregulation of genes involved in shoot repression (*BRC1*, *MAX3*), *ARR4*, a type A-response regulator (*RR*), and auxin metabolism (*SHY2*).

## 1. Introduction

*Cymbopogon citratus* (DC.) Stapf (lemongrass) is a member of the family Poaceae (https://www.itis.gov/), which is widely distributed in the tropical and subtropical regions of Africa, Asia, and America [1]. The annual world production of lemongrass oil is around 1000 tons from an area of 16,000 ha. The crop is extensively cultivated in marginal and waste lands and also along the borders as live much. The major share of lemongrass oil produced in the world is either from *Cymbopogon flexuosus* or from *C. citratus* [2]. Lemongrass is known for its high content of essential oils, such as citral, geraniol, citronellol, citronellal, linalool, elemol, 1,8-cineole, limonene, β-caryophyllene, methyl heptanone, and geranyl acetate/formate [3,4], as well as for its applications as a medicinal supplement, insect repellent, insecticide, perfumery, and pharmaceuticals [1,5,6]. Plant micropropagation via a shoot apex culture supplemented with 6-Benzylaminopurine (BAP) has been the most effective cytokinin (CK) in grasses; little bluestem (*Schizachyrium scoparium* L.) [7], vetiver grass [8], *Cymbopogon schoenanthus* [9], switchgrass (*Panicum virgatum* L.) [10], finger millet (*Eleusine coracana* (L.) Gaertn.) [11], *Pennisetum purpureum* [12], *Pennisetum × advena* ‘Rubrum’ [13], and cereals; as well as rice [14], wheat [15], wheat, durum wheat, oat, barley, triticale [16], and sorghum [17].

We have previously reported that, when using an MS medium supplemented with BAP 10 µM, 5% sucrose, and 5 g/L gelrite^TM^, a high shoot proliferation was induced (23.3 shoots/explant) [4]. In contrast, when using the natural CK 6-(γ,γ-Dimethylallylamino)purine (2-iP) 10 µM, the expected activation of shoots was substantially decreased (8.8 shoots/explant) and rooting developed in one step [4]. In this scenario, two different molecular mechanisms were activated under the BAP and 2-iP culture media in the lemongrass tissue culture. To understand the differences between the BAP and 2-iP effects in lemongrass, we reconstructed a network using the STRING database v11.5 [18] with the genes involved in shoot proliferation/repression, stem cell maintenance, CK signaling, cell cycle, and auxin signaling, based on the *Brachypodium distachyon* homologous genes present in the *Arabidopsis thaliana* genome (Figure 1). We analyzed 13 genes that are involved in CK signaling (*AHK3*, *ARR1*, and *ARR4*), stem cell maintenance (*STM*, *REV*, and *CLV3*), cell cycle regulation (*CDKA-CYCD3* complex), auxin signaling (*ARF5*), CK metabolism (*CKX1*), shoot repression (*BRC1* and *MAX3*), and auxin signaling repressor (*SHY2*) for RT-qPCR analysis. Gene expression analysis revealed that in the BAP culture medium (CM), the genes involved in shoot proliferation, such as CK signaling (*AHK3*, *ARR1*), stem cell maintenance (*STM*, *REV*, and *CLV3*), cell cycle regulation (*CDKA*-*CYCD3* complex), auxin signaling (*ARF5*), and CK metabolism (*CKX1*), were higher upregulated. In contrast, in 2-iP CM, there was an upregulation of genes involved in shoot repression (*BRC1* and *MAX3*), auxin (*SHY2*), and CK signaling activating the A-type response regulator (*RR) ARR4*.

The goal of this study was to elucidate the molecular mechanisms that BAP uses to activate shoot proliferation and 2-iP root development in lemongrass (*C. citratus*).

## 2. Results

### 2.1. Shoot Proliferation

Lemongrass plants grown in a medium supplemented with BAP 10 µM, 5% sucrose, and 5 g/L gelrite^TM^ are consistent with the canonical effects of CKs developed 23.3 shoots per initial explant after 2 months in culture (Figure 2A).

In contrast, in a medium supplemented with 2-iP 10 µM, 5% sucrose, and 5 g/L gelrite^TM^, fewer amounts of shoots were produced (8.8 shoots) and it also developed thick, red-pigmented roots in a single step of culture (Figure 2B).

We performed histological analysis of longitudinal sections of roots from both CKs in which a higher number of axillary meristems (AMs) were observed in plants grown on BAP 10 µM compared to the 2-iP 10 µM CM (Figure 2C,D).

Several publications indicate that high levels of CK promote shoot growth and suppress root formation. However, our results suggest a different role in 2-iP inducing roots (Figure 2B). Plants grown on BAP 10 µM did not produce roots (Figure 2A). Instead, in plants grown on 2-iP 10 µM, the root phenotypes were thicker and pigmented, possibly due to the biosynthesis of anthocyanins (Figure 2B).

### 2.2. Histology of C. citratus Roots

Roots derived from 2-iP plants are red pigmented and four times thicker than control (Ctrl) lemongrass roots (Figure 3). Longitudinal sections of *C. citratus* from root tips (RT) showed a root cap (calyptra) composed by several layers of cells containing statocytes (statenchyma); root cap (RC); cortex (CX); protodermis (PD); quiescent center (QC); distal meristem (DM); pith (P); meristem zone (MZ); and elongation zone (EZ) (Figure 3B,D). The cells from the RT derived from 2-iP have nuclei (3.76 µm), nucleolus (2.42 µm), and cell area of 107.07 µm^2^ (Figure 3A,B) (Table 1). In contrast, RT cells derived from Ctrl roots contain smaller nuclei (3.2 µm), nucleolus (1.88 µm) and cell area of 63.36 µm^2^ (Figure 3C,D) (Table 1).

### 2.3. Gene Expression Analysis

We analyzed the expression of 13 genes (*AHK3*, *ARR1*, *ARR4*, *STM*, *REV*, *CLV3*, *CDKA*, *CYCD3*, *ARF5*, *SHY2*, *CKX1*, *BRC1*, and *MAX3*) (Figure 1) via RT-qPCR. A different pattern expression was observed in plants grown on BAP compared to those grown on 2-iP (Figure 2 and Figure 4), confirming that CKs modulates shoot proliferation and root development. 

#### 2.3.1. CK Signaling Genes

The expression of *AHK3* and the B-type *RR ARR1* were upregulated in BAP CM (2.8 and 3.5 folds, respectively) compared to 2-iP CM (0.4 and 0.3 folds, respectively), while upregulation of the A-type *RR*, *ARR4*, was found in 2-iP-treated plants (5 fold) compared to BAP (0.2 fold) (Figure 4). 

#### 2.3.2. Stem Cell Maintenance Genes

The analysis revealed that the levels of *STM*, *CLV3*, and *REV* were higher in BAP CM (4.7, 3.2, and 2.8 folds, respectively) compared to 2-iP CM (0.2, 0.3, and 0.4 folds, respectively) (Figure 4). 

#### 2.3.3. Cell Cycle Regulation Genes

The *CDKA-CYCD3* complex remained upregulated in the BAP CM (2.1 and 1.6 folds, respectively) compared to the 2-iP CM (0.5 and 0.6 folds, respectively) (Figure 4). 

#### 2.3.4. Auxin Signaling Gene

The *ARF5* expression was revealed to be the most highly upregulated gene in BAP CM (5.4 fold) compared to the treatment with 2-iP (0.2 fold) (Figure 4). 

#### 2.3.5. Auxin Signaling Repressor Gene

The analysis reveals that *SHY2* was the most upregulated gene in 2-iP CM (3.5 fold) compared to the treatment with BAP (0.3 fold) (Figure 4). 

#### 2.3.6. CK Metabolism Gene

The *CKX1* expression remains upregulated in BAP CM (3.2 fold) compared to the CM with 2-iP (0.3 fold) (Figure 4). 

#### 2.3.7. Shoot Proliferation Repressor Genes

The analysis revealed that the levels of *BRC1* and *MAX3* remained elevated in the 2-iP CM (1.7 and 1.8 folds, respectively) when compared to the BAP CM (0.6 and 0.6 folds, respectively) (Figure 4).

## 3. Discussion

CKs have been mainly used in plant micropropagation, including Poaceae members like cereals and grasses, due to the positive regulation of shoot and the negative regulation of root development [19,20,21].

The known molecular mechanisms of CKs (zeatin, 2-iP and BAP) rely on the two-component CK signaling [22,23,24,25], cell cycle regulation [26], direct binding with type-B *RRs* [27], RNA, proteins, ribosomal proteins, and polyribosomes synthesis [28,29,30,31,32,33,34]. Furthermore, CKs are inhibitors of senescence by downregulating several transcription factors (TFs) [35] involved in several types of stress (drought, osmotic, salt, temperature, and oxidative) and crosstalk during plant stress with ABA and jasmonic acid.

Based on that, we aimed to understand by using a PPI network STRINGv11.5 database containing the molecular mechanisms involved in lemongrass grown in the high shoot proliferation medium (BAP 10 µM) versus the low shoot proliferation and root development medium (2-iP 10 µM) (Figure 2 and Figure 5).

The interpretation of the molecular mechanism found in this work is as follows:

### 3.1. Cytokinin Signaling

Interestingly, we found that different molecular mechanisms between BAP and 2-iP relied on CK signaling, specifically with the type of *RRs*. Type-A *RRs* evolved with land plants, indicating that their signaling pathway and negative feedback regulation were established simultaneously to activate vascularization and root development, whereas type-B *RRs* were already present, advocating that they had CK-independent functions [36].

In our analysis, *AHK3* and *ARR1* were higher expressed in BAP than the 2-iP-containing medium and in agreement with our results, where BAP has been demonstrated to be involved in the expression/mutant restitution of type-B *RRs* that function as repressors of root development [27,37,38,39,40,41,42,43] (Figure 4 and Figure 5).

Type-B *PeRR12* is a negative regulator of root development in poplar, repressing the *WUSCHEL*-related homeobox *PeWOX5*, which is involved in the maintenance of the stem cells in the root apical meristem (RAM), and *PeWOX11*, which interacts with the auxin signaling pathway regulating the founder cells during de novo root regeneration [44]. *PeRR12* also represses *PePIN1* and *PePIN3* transcripts, two auxin efflux carriers involved in root and shoot development [45].

Type-B *PtRR13* is a negative regulator of root development in *Populus*. A transcriptome analysis showed that it altered the expression levels of *RING1*, a negative regulator of vascularization, *PDR9*, an auxin efflux transporter, and also two apetala/ethylene responsive factors [46].

In our analysis *AHK3* was expressed lower and *ARR4* higher in the 2-iP-containing medium. 2-iP has demonstrated to be involved in the expression of type-A *RRs* that function as activators of root development (Figure 4 and Figure 5). Roots developed in 2-iP were pigmented and four times thicker than normal lemongrass roots. The histological analysis from RT revealed that they contain a prominent root cap (RC) composed by several layers of cells containing statocytes (statenchyma), CX, PD, QC, DM, P, MZ, and EZ.

When cultured in a 2,4-D-containing medium, the explants of roots from 2-iP were capable of developing the somatic embryogenesis process while in normal roots, it was not possible.

However, the BAP-containing medium activates B-type *RR*s, TFs involved in shoot proliferation, and root inhibition.

Type-A *RRs* contribute to the correct function of RAM, regulating post-transcriptionally *PIN* auxin efflux carriers. Transcriptome analysis of *Arabidopsis*-treated plants with 2-iP revealed the upregulation of genes involved in photosynthesis, ribosome biogenesis, CK responsive factors, several type-A *RRs* (*ARR3*,-*4*,-*6*,-*7*,-*8*,-*9*,-*15*,-*17*), while senescence and catabolism genes were downregulated [35].

The overexpression of type-A *RRs* in *Arabidopsis* substantially increased lateral root development and inhibits shoot development in *ARR3*,-*5*,-*6*,-*16* and -*17* when supplemented with 2-iP, where mutated type-A *RRs* led to reduced root architecture [47]. The *RcRR1*, a type-A *RRs* of *Rosa canina*, has been found to be involved in CK-modulated rhizoid organogenesis [48].

Successful rooting from shoots using 2-iP has been observed in date palm [49], amaryllis [50], sweet cherry [51], and apple [52,53]. Moreover, 2-iP is involved in nitrogen signaling and regulates root architecture and development by modulating polar auxin transport and vascular patterning in the root meristem [54].

CK signaling loss correlates with a reduced meristem size, whereas enhanced cytokinin action stimulates meristem activity. The shoot architecture is normally promoted in the stem cells located in the central zone of the shoot apical meristem (SAM) [55,56,57].

Accordingly, *STM*, *CLV3*, and *REV* genes involved in stem cell maintenance, AMs initiation, and shoot proliferation regulation were upregulated in BAP compared to 2-iP-treated shoots. High levels of the expression of *STM* activates stem cell maintenance in plant meristems [58,59,60,61,62], CK biosynthesis [63,64], meristem initiation, and AMs development [65,66,67] (Figure 5).

*CLV3* is involved in maintaining the stem cell population in the SAM and is central to continuing shoot growth [68]. The TF *REV* gene regulates the *STM* expression [67] and is an activator of shoot proliferation, where *REV* mutants develop fewer shoots [69].

### 3.2. Cell Cycle

The CK BAP induces the *CYCD3* expression at the G_1_-S cell cycle phase transition, showing that *CYCD3* induction may be a direct response to BAP or an indirect response of cells reaching a particular position in the cell cycle under the influence of BAP [70], where the same occurred in our results with BAP at the concentration of 10 μM (Figure 4 and Figure 5). *CYCD3*-*1* forms a complex with *CDKA*, which is required for the maintenance of undifferentiated cells in the SAM [71,72], determines the cell number and expansion in developing lateral organs (shoots), and mediates CK effects in apical growth and development [73] (Figure 5).

### 3.3. BRC1 Shoot Repressor

In our results, we found that BAP activates poorly *BRC1* compared to 2-iP (Figure 4 and Figure 5), as described by Xu et al. [74], in which the formation of mutant phenotypes in *Panicum virgatum* L. is rescued via the BAP application. In another example, the work of Lewis et al. [75] showed that the overexpression of *BRC1* results in a reduced number of shoots. *BRC1* encodes a shoot growth inhibitory TF [76], which in 2-iP (10 μM), its relative expression was elevated compared to BAP (10 μM) and the associated plant phenotypes show fewer AMs, but with developed roots (Figure 2).

### 3.4. SHY2 (IAA3/Short Hypocotyl 2)

CKs cooperate with other plant growth regulators to regulate root meristem development; for example, the *SHY2* gene is an essential axis of the interaction between CKs, auxin, and brassinosteroids (BR) [77,78]; it is therefore upregulated after the CM with 2-iP (Figure 5).

The expression of *SHY2* upon 2-iP CM negatively regulates polar auxin transport carried out via PIN, redistributes the auxin concentration, and induces cell proliferation. Alternatively, auxin and CK are regulated via antagonism in the meristematic and differentiation zones of the roots to control the meristem size [79].

### 3.5. Strigolactone (SL) Biosynthesis

*MAX3* and *BRC1*, which are involved in SL biosynthesis and repress shoot proliferation, were upregulated in 2-iP CM (Figure 4 and Figure 5). The exogenous supply of SL can upregulate the *BRC1* expression independently of any new protein synthesis, suggesting that *BRC1* is the most direct target of SL signaling [80,81]. *MAX3* transcript profiling revealed that the upregulation of SL biosynthetic genes was indirectly triggered by CKs and led to the activation of the *BRC1* expression [82].

### 3.6. CK Oxidase/Dehydrogenase

Exposure to 10 μM BAP caused the induction of the CK oxidase/dehydrogenase gene *CKX1*, which was also reported in the work of Vylíčilová et al. [83], suggesting that BAP has an independent mode of action or may act as a negative regulator of CK perception by upregulating specific genes, such as *CKX1* to balance the endogenous CK concentration (Figure 4 and Figure 5).

### 3.7. Auxin Response Factor

*ARF5* is an auxin response factor that regulates auxin-responsive elements (Figure 5). It plays a crucial role in plant development, especially in embryonic development, cell differentiation, and organogenesis [84]. Mutations result in a defective embryo and SAM development [84].

## 4. Materials and Methods

### 4.1. Shoot Proliferation of Lemongrass (C. citratus)

*C. citratus* plants were cultivated under standard in vitro conditions under a 16/8 h light/dark photoperiod, according to Camas-Reyes et al. [4]. Two week-old germinated seedlings of lemongrass (*C. citratus*) on an MS medium [85] supplemented with 1% sucrose, at pH 5.8 were subcultured in two different CM: an MS medium supplemented with BAP 10 µM, 5% sucrose, 5 g/L gelrite^TM^, at pH 5.8, and an MS medium supplemented with 2-iP 10 µM, 5% sucrose, 5 g/L gelrite^TM^, at pH 5.8.

### 4.2. Histology of C. citratus Shoots and Roots

Shoots and roots of *C. citratus* were randomly selected (10 samples of BAP CM and 10 samples of 2-iP CM for shoot histology; 10 samples of 2-iP CM and 10 samples of control CM for root histology). These were fixed in FAE (5% formaldehyde, 10% acetic acid, and 50% ethanol), followed by dehydration in a series of ethanol dilutions (20%, 40%, 60%, 80%, and 100% ethanol) for 2 h each. Shoot samples were embedded in Technovit 7100 (Heraeus Kulzer, Hesse-Hanau, Deu) and root samples in a mix of acetonitrile/Spurr EMS (Hatfield, UK) following the manufacturer’s instructions. Shoot sections (14 μm) were obtained on a rotary microtome (Reichert-Jung 2040, Leica (Hernalser Hauptstrasse, Vienna); and the root semithin sections (500 nm) with a Leica EM UC7 ultramicrotome Leica (Hernalser Hauptstrasse, Vienna). Tissue sections were stained with a 0.02% toluidine blue solution (HYCEL, Zapopan, Mexico) for 3 min, washed with distilled water for 1 min, and then air dried [86]. Shoot images were taken with a DM6000B microscope (Leica) and root photos were taken with a DMi8 inverted microscope (Leica). The open source image processing software FIJI (Fiji Is Just ImageJ), based on ImageJ2 [87], was used.

### 4.3. Isolation of RNA and Real Time Gene Expression Analysis

Total RNA was isolated from the shoots derived from cultures containing BAP and/or 2-iP using Trizol (Invitrogen, Carlsbad, CA, USA). The RNA concentration was measured using its absorbance at 260 nm, where a ratio of 260 nm/280 nm was assessed and its integrity was confirmed via electrophoresis in agarose 1% (*w*/*v*) gels. The samples of cDNA were amplified via RT-qPCR using Maxima SYBR Green/ROX qPCR (Thermo Scientific, Waltham, MA, USA) in a Real-Time PCR System (CFX96 BioRad). Elongation factor 1α (*EF1α*), glyceraldehyde-3-phosphate dehydrogenase (*GAPDH*), and actin (*ACT*) were used as reference genes for qPCR normalization [88] (Table 2).

Retfinder, NormFinder, Bestkeeper, and Delta-*C*t analyses were used. Three replicates were created for each of these reference genes, where the relative expression and weighted *C*t. were calculated. Next, a Delta-*C*t of each gene and the relative amount of target gene expression were analyzed using the 2^−ΔΔ*C*t^ method [89]. RT-qPCR analysis was based on at least three biological replicates with three technical repetitions for each sample and the control treatment.

To select the genes, a PPI network with a medium confidence (0.4) was performed with STRING (v11.5, http://string-db.org), based on the *B. distachyon* homologous genes present in the *A. thaliana* genome (Figure 1). The selected genes to be followed by RT-qPCR were *AHK3* (Histidine kinase 3); *ARR1* (Response regulator 1); *ARR4* (Response regulator 4); *STM* (Shoot meristemless); *REV* (Revoluta); *CLV3* (Clavata 3); *CDKA* (Cyclin-dependent kinase A1); *CYCD3* (Cyclin D3); *ARF5* (Monopteros); *SHY2* (Short hypocotyl 2); *CKX1* (CK oxidase/dehydrogenase 1); *BRC1* (Branched 1); and *MAX3* (Carotenoid cleavage dioxygenase 7, CCD7) (Table 2).

The gene identifier was made according to UniProt (http://www.uniprot.org) and NCBI (http://www.ncbi.nlm.nih.gov) databases. Sequences of *B. distachyon* genes were analyzed using blastN and blastP. Oligonucleotides were designed for qPCR (2^−ΔΔ*C*t^ method analysis) gene expression or transcriptional analysis.

## 5. Conclusions and Perspectives

In lemongrass (*C. citratus*), shoots developed in the BAP-containing medium drive the mechanisms to activate gene expression for AMs; stem cell maintenance mediated by *STM*, *REV*, and *CLV3*; CK metabolism *CKX1*; CK signaling mediated by *AHK3*, *ARR1*; cell cycle progression by *CDKA-CYCD3*; and auxin signaling by *ARF5.*

Plants developed in 2-iP showed higher expression levels in the genes involved in root development; *MAX3*; *BRC1*, a major shoot repressor; *SHY2* and *ARR4*.

The molecular mechanisms of BAP activating *ARR1* (B-type *RRs)* to produce shoots and repress root development are in agreement with the high shoot proliferation in *C. citratus.*

The molecular mechanisms of 2-iP activating *ARR4* (A-type *RRs*) to produce roots and repress shoot proliferation are in agreement with the 2-iP phenotype showed in *C. citratus.*

Further experiments on transcriptomic/proteomic/metabolomics analysis are required to satisfactorily corroborate our findings.

## Figures and Tables

**Figure 1 plants-12-03637-f001:**
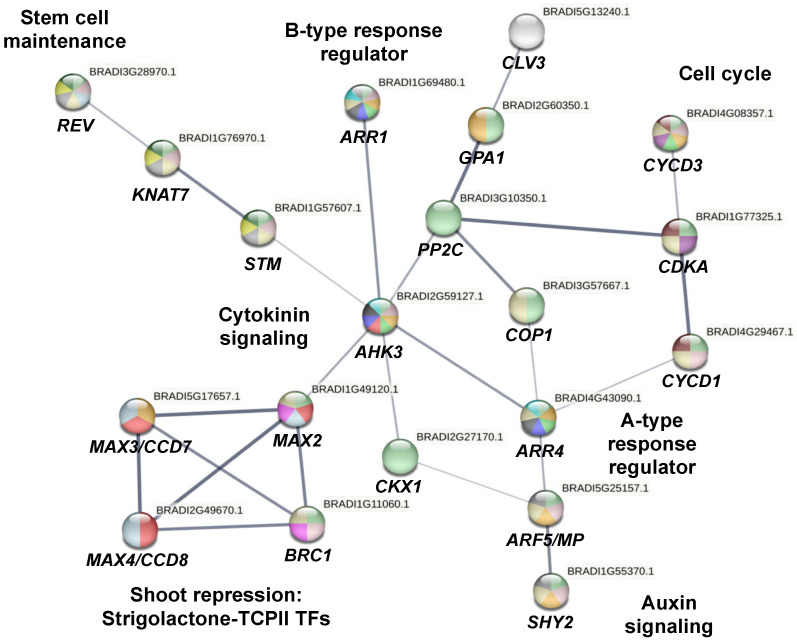
Protein–Protein interaction (PPI) network of genes involved in cytokinin (CK) signaling, stem cell maintenance, cell cycle regulation, auxin signaling, auxin repression, CK metabolism, and shoot proliferation/repressor in *C. citratus*. This PPI network was reconstructed from a gene network with an average confidence of 0.4 in STRING (https://string-db.org/, v11.5).

**Figure 2 plants-12-03637-f002:**
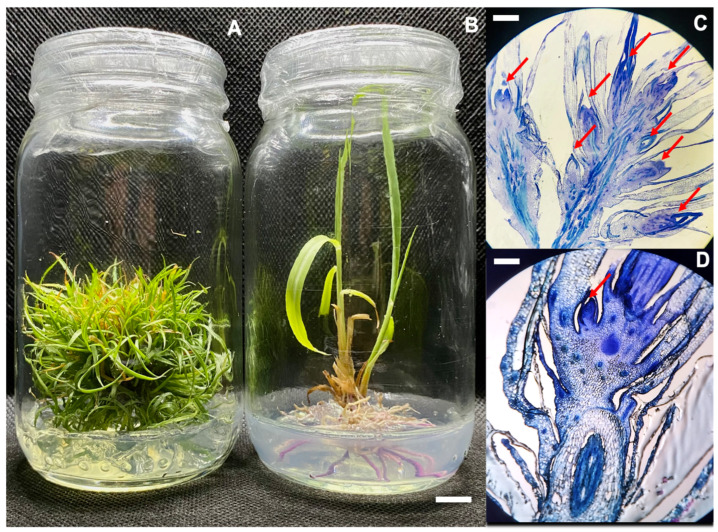
Phenotypes of lemongrass plants (*C. citratus*) grown on medium containing BAP 10 µM, 5% sucrose and 5 g/L gelrite^TM^ (**A**), compared with plants grown in medium with 2-iP 10 µM, 5% sucrose and 5 g/L gelrite^TM^ (**B**) (Scale bar = 1 cm). Longitudinal section of axillary meristems (AMs) grown on medium with BAP 10 µM (**C**) compared with sections derived from plants grown in medium with 2-iP 10 µM (**D**). Red arrows indicate AMs (Scale bar = 0.7 cm).

**Figure 3 plants-12-03637-f003:**
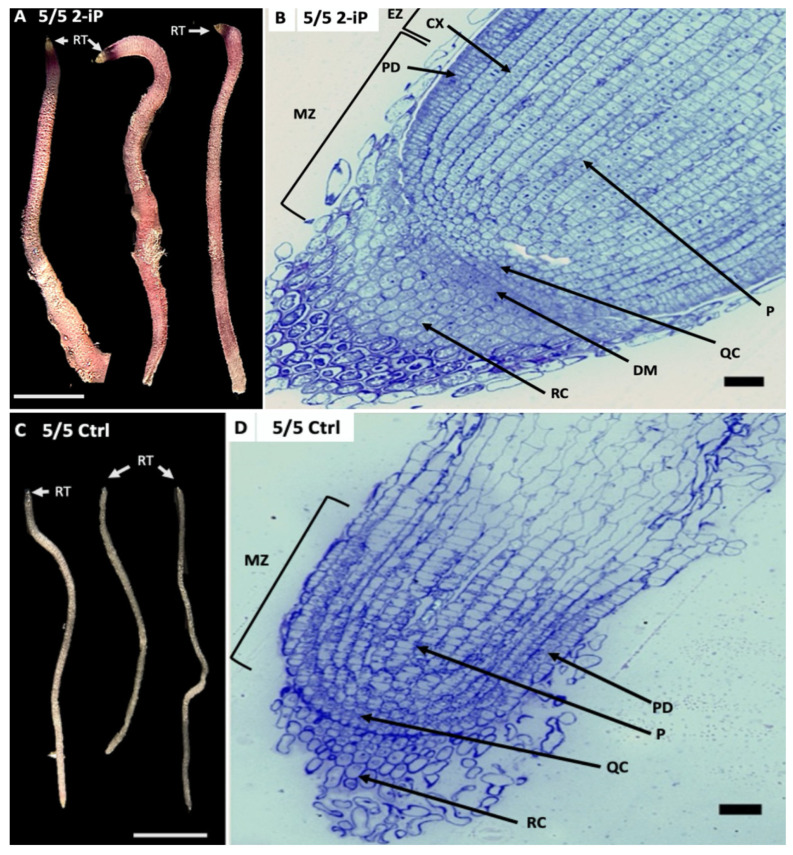
Histological analysis of longitudinal sections of *C. citratus* root tips (RT) developed in 2-iP (**A**,**B**) and control (Ctrl) medium without hormones (**C**,**D**). (**A**) Roots developed in MS medium supplemented with 2-iP 10 µM, 5% sucrose and 5 g/L gelrite^TM^ (5/5 2-iP) culture medium (Scale bar = 4 mm). (**B**) Longitudinal sections of 1 µm of RT from 5/5 2-iP. A prominent root cap (RC) composed by several layers of cells containing statocytes (statenchyma), cortex (CX), protodermis (PD), quiescent center (QC), distal meristem (DM), pith (P), meristem zone (MZ), and elongation zone (EZ) were observed. Longitudinal layers of cells with prominent nuclei were observed (Scale bar = 10 μm). (**C**) Roots developed from plants without hormones in MS medium supplemented with 5% sucrose and 5 g/L gelrite^TM^ (Scale bar = 4 mm). (**D**) Longitudinal sections of 1 µm of RT from Ctrl plants without hormones. The MZ is composed by a few layers of cells containing QC, PD, and P. It was not possible to visualize the CX, DM, and EZ (Scale bar = 10 μm).

**Figure 4 plants-12-03637-f004:**
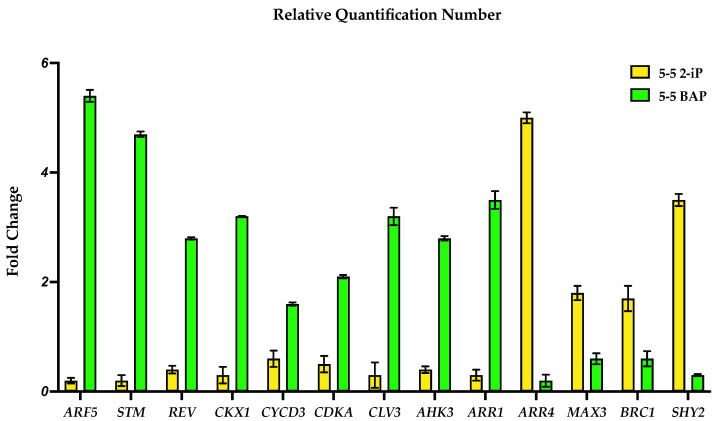
Relative gene expression levels of 13 regulatory genes present in *C. citratus* plants grown on BAP 10 µM, 5% sucrose and 5 g/L gelrite^TM^ compared to plants grown in medium with 2-iP 10 µM, 5% sucrose, and 5 g/L gelrite^TM^.

**Figure 5 plants-12-03637-f005:**
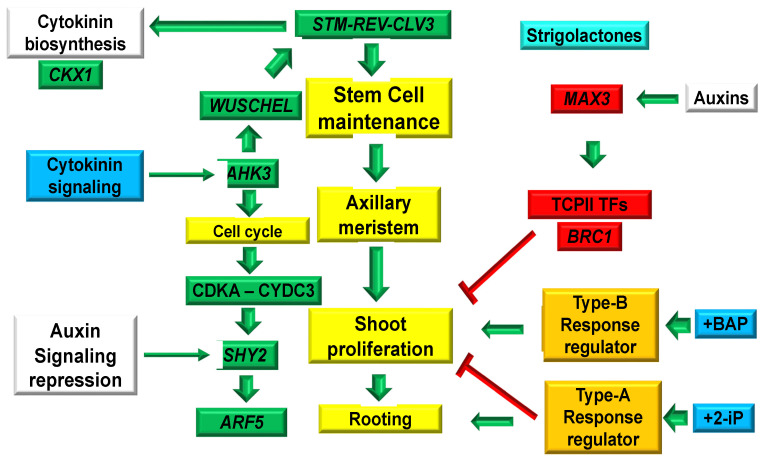
Molecular mechanisms describing proliferation and inhibition of shoots and root development in lemongrass (*C. citratus*) under BAP- and 2-iP-containing medium. Red indicators represent inhibition effects, and the green arrows represent activations. Green elements are the components of protein interactions, yellow elements represent biological functions, blue elements represent plant hormones, red elements are branching repressors, white elements represent auxin influence, orange elements represent the activation of type-A and -B response regulators (*RRs*) via CK signaling, and light blue element represents strigolactone (SL).

**Table 1 plants-12-03637-t001:** Nuclei and nucleolus size from root-tip sections of plants derived from 2-iP and controls without hormones.

	Culture Medium	Culture Medium	
	5/5 2-iP (±SD)	5/5 Ctrl (±SD)	*p*-Value
**Nucleus (** **μ** **m)**	3.76 ± 0.23	3.2 ± 0.40	0.023 *
**Nucleolus (** **μ** **m)**	2.42 ± 0.38	1.88 ± 0.24	0.0001 **
**Cell area (** **μ** **m^2^)**	107.07 ± 20.91	63.36 ± 30.91	0.0001 **

Morphological differences between organelles of roots of *C. citratus* regenerated from medium containing 2-iP 10 μM, 5% sucrose and 5 g/L gelrite™ (5/5 2-iP) and medium control (Ctrl) containing 5% sucrose and 5 g/L gelrite™ (5/5 Ctrl) after six months of culture. Sizes Nucleus (*p*-value = < 0.023 *); Nucleolus (*p*-value < 0.0001 **), and Cell area (*p*-value < 0.0001 **). Differences in variables among treatments were tested using factorial analysis of variance (ANOVA). Values are means ± standard deviation and Software ImageJ2. * (*p* < 0.05) non-significant different.; ** (*p* > 0.0001) significant different.

**Table 2 plants-12-03637-t002:** Primer design of genes that were analyzed during shoot and root induction of lemongrass (*C. citratus*).

ID	String	Function	Forward	Reverse
BRADI1G57607.1	*STM*	Stem cell maintenance	TGCACTACAAGTGGCCTTAC	CCGTTTCCTCTGGTTGATG
BRADI3G28970.1	*REV*	Stem cell maintenance	CTAGTTCCTGCACGGGATTT	CACCTCCTGAACCACTCAAA
NM_001124926.2	*CLV3*	Stem cell maintenance	CATGATGCTTCTGATCTCAC	GGGAGCTGAAAGTTGTTTC
NP_001388401.1	*CKX1*	CK metabolism	CTGATCGCCGCGCTGATCGC	CCCAGGACGGCGTCGAACAG
XM_015794058.2	*AHK3*	CK signaling	GTGAGGGGGAGCCTGGTGGCG	CTTCTTGATCGCCCACCCTTG
XM_003558408.4	*ARR1*	CK signaling	GATTATCCGAGAGGTGCGGG	TGGACGAGCATCTGCTTGAG
XM_010240442.3	*ARR4*	CK signaling	AGGCTCCTCAAGACCTCTTCT	CACGTTCACCTCAACATCCTGC
BRADI1G77325.1	*CDKA*	Cell cycle regulation	CTTCTTGGAGCAAGGCAGTAT	TCTCAGAATCTCCAGGGAACA
BRADI4G08357.1	*CYCD3*	Cell cycle regulation	TCGCTGACTCGCTCTACT	ACATGAGCTCCTCCTCCTT
XM_003557426.3	*ARF5*	Auxin signaling	CTGCGCCTCGGCCCTCCTGG	CATATCCCATGGCACGTCTCC
XM_003565937.4	*SHY2*	Auxin repressor	GCCGCCGGCGGCCGCGCGGAC	CTCGTCCTTCTCCTCGTACGC
XM_014900703.2	*BRC1*	Shoot repression	AGGAGTTGCGGAGAAGTTG	CGAGATGATGAGCAACGACA
XM_003581453.4	*MAX3*	Shoot repression	GTGGCCAACACGAGCGTGCTC	CCGAACATCCTGTGGAAATG
*	*EF1α*		TCTCGGAGCTGCTCACCAA	GTCGCCATTCTTGAGGAACTTG
*	*GAPDH*		CCCGACGAGCCCATCAT	CTTTTGGTCGAGCACCTTGAC
*	*ACT*		GACTACGACCAGGAGATGGAGACT	ATGACCTGTCCATCAGGAAGCT

* The sequences of the reference genes used were obtained from Meena et al. [88].

## Data Availability

The datasets supporting the conclusions of this article are included within the article.

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
