# Peer review of "The Cytokinins BAP and 2-iP Modulate Different Molecular Mechanisms on Shoot Proliferation and Root Development in Lemongrass (Cymbopogon citratus)"

_plants, 2023, doi:10.3390/plants12203637_

Round 1

Reviewer 1 Report

The manuscript (ID plants-2629192) submitted for review in Plants journal is within its scientific scope. The work is well written and I read it with pleasure. The results are interesting and well presented and analyzed. I have no comments regarding the methodological part of the manuscript. To sum up, in my opinion the work is ready for acceptance. Congratulations to the authors. Good job!

Author Response

Dear reviewer 1, thank you very much for reading the present manuscript.

Reviewer 2 Report

This study makes a favorable impression because, in addition to cultural methods, it used methods for analyzing the expression of associated genes.

The abstract reflects the results obtained, although it would be desirable to reformulate the sentence of Lines 21-24, since it is poorly understood.

Introduction

It is advisable to mention similar studies conducted on other monocots, especially cultivated cereals (wheat, barley, etc.). At the end of the Introduction, it is necessary to clearly indicate what goals the researchers set and what they wanted to find out.  For example: “The goal of the study was…” It is better to simply move the statement of the work done into a discussion or conclusion.

Results

Histology of C. citratus roots

In Figure 3, Figures 3c and 3d give a strange impression. Figure 3c may be more likely to be the norm (without hormone) than 3d. Are the authors confused?

Table 1. Looks unconvincing, since the authors do not provide photos of the cells on the example of which they calculated the parameters of cells and subcellular compartments. It is necessary to provide a photo along with scale bars. There is no indication from which root zone the cells were examined. Meristem? Elongation zone? The strange term “height” is used repeatedly. Do the authors mean “length”? In the table, in the organelle graph, cell width and height are collected together. However, these are not organelles. There is no indication of statistical aspects below the table.

In the Discussion, the results of histological analysis are not discussed at all.

Author Response

Responses to Reviewer 2:  Round 2
1.- At the end of the Introduction section. The sentence: “Interestingly, the differences between...” (Lines 73-74) is completely inappropriate there. It is better to move it to the Discussion section.
The sentence was removed from the introduction transferred to the discussion
The interpretation of the molecular mechanism found in this work is as follows:
3.1 Cytokinin signaling. Interestingly, we found that different molecular mechanisms between BAP and 2-iP relied on the CK signaling, specifically with the type of response regulators (RRs).
2.- Section Materials and methods. The BAP and 2-iP application methodology is poorly described. It is necessary to provide the composition of the media, and more specifically, the content of growth regulators in this section. 
Two weeks-old germinated seedlings of lemongrass C. citratus on MS medium [85], supplemented with 1% sucrose, pH 5.8 were subcultured in two different CM: MS medium supplemented with BAP 10 µM, 5% sucrose, 5 g/L gelrite, pH 5.8 and MS medium supplemented with with 2-iP 10 µM, 5% sucrose, 5 g/L gelrite, pH 5.8.
For what period of time were the plants cultivated on these media?
Plants were subcultures monthly, and after 6 rounds they were used for gene expression analysis.
 More precisely, when were the roots cut out for analysis?
Roots and shoots were dissected after 6 months of culture, and then immediately were frozen for RNA extraction and further gene expression analysis. 
Next, the term "treatment". It is rather applicable when the test substance is introduced from the outside. Here, the test substances were initially contained in the medium. Therefore, more precise use of terminology is necessary.
We have changed the term treatment for culture medium (CM).
Table 1. The title matches the essence of the table, but the name "organelle" only applies to the nucleus. And the “cell area” is in no way an organelle.

Caption for Figure 3. What is "5/5...medium"? This needs to be explained, for example, in the Materials and Methods section
Roots developed in MS medium supplemented with 5% sucrose and 5 g/L gelrite (5/5 2-iP) culture medium.
This manuscript requires further improvement

Responses to Reviewer 2:

I have attended your comments to the manuscript point by point that you kindly asked. Our manuscript is entitled:

The cytokinins BAP and 2-iP modulate different molecular mechanisms on shoot proliferation and root development in lemongrass (Cymbopogon citratus)

Authored by:

Maria del Rosario Cárdenas-Aquino , Alberto Camas-Reyes , Eliana Valencia-Lozano , Lorena López Sánchez , Agustino Martinez-Antonio * , José Luis Cabrera-Ponce *

Answers to Reviewer 2

This study makes a favorable impression because, in addition to cultural methods, it used methods for analyzing the expression of associated genes.

1.- The abstract reflects the results obtained, although it would be desirable to reformulate the sentence of Lines 21-24, since it is poorly understood.

A different pattern of gene expression was observed between BAP versus 2-iP treated plants.  In shoots derived from BAP we found upregulated genes that have already been demonstrated to be involved in de novo shoot proliferation development in several plants, CK receptors (AHK3, ARR1), stem cell maintenance (STM, REV and CLV3), cell cycle regulation (CDKA-CYCD3 complex), as well as auxin response factor (ARF5) and CK metabolism (CKX1). In contrast, in the 2-iP treatment, there was an upregulation of genes involved in shoot repression (BRC1, MAX3), ARR4 type A-response regulator and the auxin metabolism (SHY2).

Introduction

2.- It is advisable to mention similar studies conducted on other monocots, especially cultivated cereals (wheat, barley, etc.).

It was corrected:

We added 11 new references

Plant micropropagation through shoot apex culture supplemented with 6-Benzylaminopurine (BAP) has been the most effective cytokinin (CK) in grasses; Little Bluestem (Schizachyrium scoparium L.) (Hawkins et al. 2019),Vetiver grass (Sompornpailin et al. 2016), Cymbopogon schoenanthus (Abdelsalam et al. 2017), Switchgrass (Panicum virgatum L.), (Alexandrova et al 1996), finger millet (Eleusine coracana (L.) Gaertn.) (Satish et al. 2015), Pennisetum purpureum (Karaswa et al. 2001), Pennisetum × advena ‘Rubrum’ (Pożoga et al. 2023) and cereals; Rice (Nautiyal et al. 2022), Wheat (Tanzarella et al. 1985), Wheat, Durum Wheat, Oat, Barley, Triticale (Ganeshan et al. 2006), Sorghum (Pola et al. 2007).

3.- At the end of the Introduction, it is necessary to clearly indicate what goals the researchers set and what they wanted to find out.

The goal of this study is to elucidate the molecular mechanisms that BAP use to activate shoot proliferation and 2-iP root development in Cymbopogon C. citratus. Interestingly, the differences between both hormones can be explained according to the action of A and B-type response regulators involved in CK signaling, leading to a different development program.

Results

Histology of C. citratus roots

4.- In Figure 3, Figures 3c and 3d give a strange impression. Figure 3c may be more likely to be the norm (without hormone) than 3d. Are the authors confused?

We are not confused, the histological analysis of figure 3C and D correspond to the same medium without hormones, since in medium containing BAP, roots were not developed.

The figure 3 was modified according to your questionary.

5.- Table 1. Looks unconvincing, since the authors do not provide photos of the cells on the example of which they calculated the parameters of cells and subcellular compartments. It is necessary to provide a photo along with scale bars.

Table 1 was changed. Standard deviation was added

Table 1. Size of the organelles in the longitudinal section of root apices.

Treatment

Treatment

Organelle

5/5 2-iP (±SD)

5/5 Ctrl (±SD)

P-value

Nucleus (mm)

3.76±0.23

3.2±0.40

0.023*

Nucleolus (mm)

2.42±0.38

1.88±0.24

0.0001**

Cell area (mm2)

107.07±20.91

63.36±30.91

0.0001**

Morphological differences between organelles of roots of C. citratus regenerated from 5/5 2-iP and 5/5 Ctrl medium after six months. Sizes Nucleus p- value = < 0.023*; Nucleolus p value <0.0001**), and Cell area p value <0.0001**). Differences in variables among treatments were tested using factorial analysis of variance (ANOVA). Values are means ± standard deviation and Software ImageJ. *(P < 0.05) non-significant different.; ** (P>0.00) significant different.

6.- There is no indication from which root zone the cells were examined. Meristem? Elongation zone?

The root analysis was made in the root tip containingt the root cap, the quiescent center and the meristem zone and distal meristem.

7.- The strange term “height” is used repeatedly. Do the authors mean “length”?

It was changed for length.

8.- In the table, in the organelle graph, cell width and height are collected together. However, these are not organelles. There is no indication of statistical aspects below the table.

An statistical analysis was added to the evaluation.

9.- In the Discussion, the results of histological analysis are not discussed at all.

It was added in the discussion.

Reviewer 3 Report

In this manuscript, authors did a study about cytokinins BAP and 2-iP modulate different molecular mechanisms on shoot proliferation and root development in lemongrass (Cymbopogon citratus). I think this study is meaningful and the research logic is OK. The data is enough to support the result. However, there are some critical issues should be clarified as follows:

1. In the part of abstract, I suggest authors delete this setence “The molecular mechanisms that both regulators BAP and 2-iP induced in shoot and root proliferation will be discussed in this manuscript. ”

2. The annual output of Cymbopogon citratus (DC.) should be provided in the whole world or in America or in area.

3. The data in Table 1 should be given standard deviation at the same time.

4. In the part of discussion, I dont think it is necessary to separate so many paragraphs.

5.  I also dont think authors give a very good discussion for the whole study logic. Please rearrange the discussion, its better not to separate different parts for discussion.

6. In the part of materials and methods, the number of samples were not enough.

7. The quality of writing is pretty poor, discussion is too weak and conclusion is lack of focus.

8. The whole paper is too scattered to form a logic whole story.

Author Response

Answers to Reviewer 3

Comments and Suggestions for Authors

In this manuscript, authors did a study about cytokinins BAP and 2-iP modulate different molecular mechanisms on shoot proliferation and root development in lemongrass (Cymbopogon citratus). I think this study is meaningful and the research logic is OK. The data is enough to support the result. However, there are some critical issues should be clarified as follows:

 Point by point corrections

1.- In the part of abstract, I suggest authors delete this setence “The molecular mechanisms that both regulators BAP and 2-iP induced in shoot and root proliferation will be discussed in this manuscript. ”

The sentence was eliminated from the abstract.

2.- The annual output of Cymbopogon citratus (DC.) should be provided in the whole world or in America or in area.

It was corrected.

3.- The data in Table 1 should be given standard deviation at the same time.

The table 1 was corrected.

Table 1. Size of the organelles in the longitudinal section of root apices.

Treatment

Treatment

Organelle

5/5 2-iP (±SD)

5/5 Ctrl (±SD)

P-value

Nucleus (mm)

3.76±0.23

3.2±0.40

0.023*

Nucleolus (mm)

2.42±0.38

1.88±0.24

0.0001**

Cell area (mm2)

107.07±20.91

63.36±30.91

0.0001**

4.- In the part of discussion, I don’t think it is necessary to separate so many paragraphs.

It was modified, and better integrated.

  1. I also don’t think authors give a very good discussion for the whole study logic. Please rearrange the discussion, it’s better not to separate different parts for discussion.

Please and apology for this.  What dou you mean with LOGIC?.

Because, If you mean that Logic is: “Reasoning conducted or assessed according to strict principles of validity”. That is in fact, what we are trying to do in this manuscript.

Didn’t you understand the molecular mechanisms that 2-iP and BAP induced differentially in lemongrass?.

  1. In the part of materials and methods, the number of samples were not enough.

It was corrected.

  1. The quality of writing is pretty poor, discussion is too weak and conclusion is lack of focus.

Dear reviewer 3, once again an apology, but: A weak discussion is when you have just a few scientific evidences that positive or negatively correlates with your experimental work. In the discussion, we cited more than 50 papers that explain LOGICALLY what we are explaining, many of this papers have a high impact factor.

To speculate saying: Taken all together our work is in agreement with the rest of the world, doesn’t make it powerfull.

  1. The whole paper is too scattered to form a logic whole story.

Once again, I am not in agreement with your personal point of view. Scattered is when your mind is disperseand are not able to integrate any scientific information you are trying to publish. But, when you are just trying to integrate your work, you have to use new plattforms to better explain the experiments and of course you have to study a lot, and it takes time.

Round 2

Reviewer 2 Report

The authors have improved the quality of the manuscript and have done some work. However, there are some comments remaining.

At the end of the Introduction section. The sentence: “Interestingly, the differences between...” (Lines 73-74) is completely inappropriate there. It is better to move it to the Discussion section.

Section Materials and methods. The BAP and 2-iP application methodology is poorly described. It is necessary to provide the composition of the media, and more specifically, the content of growth regulators in this section. For what period of time were the plants cultivated on these media? More precisely, when were the roots cut out for analysis? Next, the term "treatment". It is rather applicable when the test substance is introduced from the outside. Here, the test substances were initially contained in the medium. Therefore, more precise use of terminology is necessary.

Table 1. The title matches the essence of the table, but the name "organelle" only applies to the nucleus. And the “cell area” is in no way an organelle.

Caption for Figure 3. What is "5/5...medium"? This needs to be explained, for example, in the Materials and Methods section

This manuscript requires further improvement.

Author Response

Responses to Reviewer 2:  Round 2

1.- At the end of the Introduction section. The sentence: “Interestingly, the differences between...” (Lines 73-74) is completely inappropriate there. It is better to move it to the Discussion section.

The sentence was removed from the introduction transferred to the discussion

The interpretation of the molecular mechanism found in this work is as follows:

3.1 Cytokinin signaling. Interestingly, we found that different molecular mechanisms between BAP and 2-iP relied on the CK signaling, specifically with the type of response regulators (RRs).

2.- Section Materials and methods. The BAP and 2-iP application methodology is poorly described. It is necessary to provide the composition of the media, and more specifically, the content of growth regulators in this section.

Two weeks-old germinated seedlings of lemongrass C. citratus on MS medium [85], supplemented with 1% sucrose, pH 5.8 were subcultured in two different CM: MS medium supplemented with BAP 10 µM, 5% sucrose, 5 g/L gelriteä, pH 5.8 and MS medium supplemented with with 2-iP 10 µM, 5% sucrose, 5 g/L gelriteä, pH 5.8.

For what period of time were the plants cultivated on these media?

Plants were subcultures monthly, and after 6 rounds they were used for gene expression analysis.

 More precisely, when were the roots cut out for analysis?

Roots and shoots were dissected after 6 months of culture, and then immediately were frozen for RNA extraction and further gene expression analysis.

Next, the term "treatment". It is rather applicable when the test substance is introduced from the outside. Here, the test substances were initially contained in the medium. Therefore, more precise use of terminology is necessary.

We have changed the term treatment for culture medium (CM).

Table 1. The title matches the essence of the table, but the name "organelle" only applies to the nucleus. And the “cell area” is in no way an organelle.

Caption for Figure 3. What is "5/5...medium"? This needs to be explained, for example, in the Materials and Methods section

Roots developed in MS medium supplemented with 5% sucrose and 5 g/L gelriteä (5/5 2-iP) culture medium.

This manuscript requires further improvement

Reviewer 3 Report

All issues were addressed.

Author Response

Thank you very much for your time

Round 3

Reviewer 2 Report

The authors have made the necessary changes, therefore, the article can be accepted in present form